# Screening and Optimization of Solid-State Fermentation for *Esteya vermicola*, an Entomopathogenic Fungus Against the Major Forest Pest Pine Wood Nematode

**DOI:** 10.3390/microorganisms13020434

**Published:** 2025-02-17

**Authors:** Lanwen Zhang, Yongxia Li, Xiaojian Wen, Xuan Wang, Wei Zhang, Dongzhen Li, Yuqian Feng, Zhenkai Liu, Xingyao Zhang

**Affiliations:** 1Key Laboratory of Forest Protection of National Forestry and Grassland Administration, Ecology and Nature Conservation Institute, Chinese Academy of Forestry, Beijing 100091, China; lanwen9900@163.com (L.Z.); wenxj@caf.ac.cn (X.W.); jiuwozhidao@163.com (X.W.); zhangwei1@caf.ac.cn (W.Z.); lidongzhen1949@163.com (D.L.); fengyq@caf.ac.cn (Y.F.); oklzkk@163.com (Z.L.); zhangxingyao@126.com (X.Z.); 2Co-Innovation Center for Sustainable Forestry in Southern China, Nanjing Forestry University, Nanjing 210073, China; 3Kunyushan Forest Ecosystem National Observation and Research Station, Yantai 264100, China

**Keywords:** *Esteya vermicola*, *Bursaphelenchus xylophilus*, solid-state fermentation, orthogonal experimental design, response surface methodology

## Abstract

Pine wilt disease (PWD), caused by the pine wood nematode (PWN, *Bursaphelenchus xylophilus*), is one of the most serious threats to pine forests worldwide. The fungus *Esteya vermicola*, with its lunate conidia capable of parasitizing the PWN, has shown promise as an efficient biological control agent against PWD. Solid-state fermentation (SSF) is preferred for large-scale applications in the field, as it facilitates microbial agent transport and ensures a long shelf life. However, research on enhancing the yield of lunate conidia from *E. vermicola* through SSF is limited. In this study, we initially achieved a yield of 3.04 × 10^8^ conidia/g using a basic SSF medium composed of wheat bran, corn flour, and soybean flour. To improve this yield, we employed an orthogonal experimental design (OED) to identify the optimal medium composition, which required a wheat bran-to-corn flour-to soybean flour ratio of 7:2:1 (*w*/*w*/*w*), a substrate-to-water ratio of 1:0.7 (*w*/*v*), and the addition of 1.33% (*w*/*w*) glucose, 1.33% (*w*/*w*) yeast extract fermentation, and 1.33% (*w*/*w*) MgSO_4_. Using the response surface methodology (RSM), we calculated the optimal fermentation conditions, which were 24.9 °C, 78.0% relative humidity (RH), an inoculation volume of 16.3% (*v*/*w*), and a fermentation time of 7.1 days. Under these conditions, the yield of lunate conidia reached a maximum of 16.58 × 10^8^ conidia/g, a 4.45-fold increase after optimization. This study improved the yield of *E. vermicola* lunate conidia and provides insights for developing biopesticides based on this strain.

## 1. Introduction

Pine wilt disease is primarily caused by the pine wood nematode (PWN, *Bursaphelenchus xylophilus*) and is a devastating disease that particularly affects pine trees (*Pinus* spp.) [1]. The primary vectors for the transmission of PWN are beetles of the genus *Monochamus* Dejean 1821 spp. [2]. The PWN, native to North America [3], was introduced to Japan in the early 20th century through the timber trade [4]. Since then, it has spread to East Asia, including China [5] and South Korea [6], as well as to Portugal [7] and Spain [8] in Europe. The PWN has led to significant ecological and economic losses in forests in its invasive areas, prompting increased efforts for its prevention and control. In China, common control measures for pine wilt disease include quarantine, epidemic monitoring, the removal of infected trees, and both chemical and biological control methods [9]. Chemical control primarily involves aerial spraying to manage the vector insect, the Asian longhorned beetle (*Monochamus alternatus*), as well as trunk injection to control the PWN pathogen. However, the prolonged use of chemical agents can result in resistance, environmental pollution, and negative effects on biodiversity [10]. As a result, there is a growing emphasis on developing and implementing biological control methods.

*Esteya vermicola* is an entomopathogenic fungus targeting the PWN. This fungus was first reported in 1999 as an endophytic fungus parasitizing the PWN, demonstrating high infectivity and significant potential as a biological control agent for pine wilt disease [11]. This fungus has been identified globally, with primary distributions in East Asia, Europe, and South America [12,13,14,15,16,17,18]. *E. vermicola* is characterized by the production of two types of conidia: lunate conidia and bacilloid conidia. Only the lunate conidia possess adhesive properties, allowing them to attach to the PWN. Upon germination, these conidia form penetration structures that breach the nematode’s body wall, subsequently producing hyphae that grow and extend within the nematode, causing damage to its organs and tissues, which typically results in the nematode’s death within four days [14,18]. Previous studies indicated that *E. vermicola* can survive in the resin of pine trees without causing necrosis or discoloration [19] and that it successfully infected PWNs within the inoculated trees [20]. In field trials conducted in South Korea, Ref. [21] found that over 30% of pine trees inoculated with *E. vermicola* survived for six consecutive years, whereas all the pine trees that were not inoculated with *E. vermicola* died. However, the effectiveness of *E. vermicola* in controlling PWN is related to the quantity of conidia produced. The composition of the culture medium and the cultivation conditions can significantly influence the conidia production rate and stability and the environmental resilience of the fungus in practical applications [22]. Therefore, to further investigate the application of *E. vermicola* in the field, it is essential to produce a large quantity of lunate conidia.

Currently, *E. vermicola* is primarily cultured on potato dextrose agar (PDA) medium for experimental purposes, with conidia collected after 5 to 7 days to prepare conidial suspensions [23,24]. However, this method yields a relatively low quantity of conidia. Thus, when demand is high, liquid fermentation or SSF is typically employed [25]. Liquid fermentation allows *E. vermicola* to produce a large number of conidia and mycelium within 7 days [26,27]. Nevertheless, the conidia produced through this method have thin walls, making them prone to inactivation and resulting in a short shelf life. Additionally, challenges such as high transportation costs, susceptibility to contamination, and limited shelf life are associated with liquid fermentation. In contrast, SSF can generate a substantial quantity of aerial conidia with strong environmental resilience, along with providing advantages such as convenient transportation and an extended shelf life [26,28,29,30,31,32]. SSF refers to microbial fermentation conducted on solid substrate materials that provide the necessary environment and nutrients for microbial growth and conidia production, with the fermentation substrate containing little to no free-flowing water [33,34]. The SSF substrate typically includes carbon sources (such as rice, corn, wheat bran, sugarcane bagasse, and sorghum), nitrogen sources (such as soybean flour, fish meal, and urea), minerals (such as perlite and clay particles), and additives (such as vitamins and trace elements). An appropriate amount of moisture must also be added to the substrate to support microbial growth and conidia production [35]. The combination and proportions of these components may vary depending on the specific fermentation objectives and the microbial species involved. Research has demonstrated that using rice as a substrate for the SSF of *E. vermicola* can yield a large number of conidia after a cultivation period of 9 days [32]. Additionally, mixing pine wood powder with wheat bran for the SSF of *E. vermicola* results in the highest conidial germination rate and resistance after 7 days of cultivation [26]. However, the specific formulation for the SSF of *E. vermicola* to produce lunate conidia has not yet been clearly defined. Wheat bran, corn flour, and soybean flour are ideal SSF substrates due to their rich nutritional content, suitable physical properties, low cost, and easy availability. To explore methods for the large-scale production of lunate conidia of *E. vermicola*, this study selected wheat bran, corn flour, and soybean flour as the base SSF substrates, focusing on screening the optimal proportions of the fermentation substrates and optimizing the nutritional and cultivation conditions.

## 2. Materials and Methods

### 2.1. Fungal Strains and Culture Media

*Esteya vermicola* Fxy121, isolated from the bark of Yunnan pine (*Pinus yunnanensis* Franch.) in Yunnan Province, is preserved in the Pine Wood Nematode Research Laboratory, Ecology and Nature Conservation Institute, Chinese Academy of Forestry. Before the experiment, 5 mm diameter plugs ware taken from the actively growing edges of Fxy121 that had been cultured on the medium for 2 weeks. These plugs were inoculated into the center of plates containing potato dextrose agar (PDA) medium (USA, BD), which were sealed with a film, followed by incubation at 25 °C for an additional 2 weeks. The actively growing edges were then transferred to potato dextrose broth (PDB) medium (USA, BD) and shaken at 200 r/min at 25 °C for 7 days. Subsequently, the culture was filtered through sterile gauze to obtain an Fxy121 spore suspension. The spore concentration of the suspension was determined using a hemocytometer (Bkmam, Changde, China) and was adjusted to a value of 1 × 10^8^ conidia/mL for the Fxy121 spore inoculum, which was stored at 4 °C for later use.

### 2.2. Solid-State Fermentation (SSF) Culture

For the SSF process, common substrates such as wheat bran, corn flour, and soybean meal were mixed in equal proportions, totaling 10 g, in a 250 mL Erlenmeyer flask. The substrate mixture was then combined with sterile water at a 1:1 (*w*/*v*) ratio, resulting in an initial pH of 5.0 and a moisture content of 55%. The substrate was sterilized at 121 °C for 20 min and subsequently cooled to room temperature. Following this, a 10% (*v*/*w*) suspension of Fxy121 spores was added to the cooled substrate and thoroughly mixed. The mixture was incubated at 25 °C and 75% relative humidity (RH) for 9 days to measure the yield of the conidia.

### 2.3. Screening and Optimization of Substrate and Supplementary Nutrients

To determine the optimal SSF substrate for Fxy121, an orthogonal experimental design (OED) was employed. This design included four factors: wheat bran, corn flour, soybean flour, and the ratio of substrate to water, with each factor tested at five different levels (Appendix A). After 9 days of cultivation, the best substrate combination was selected based on the yield of lunate conidia.

In the Fxy121 solid fermentation medium, various carbon sources at a concentration of 2% (*w*/*w*), including glucose (Shanghai Aladdin, Shanghai, China), sucrose (China, Shanghai Aladdin), lactose (China, Shanghai Aladdin), maltose (China, Shanghai Aladdin), soluble starch (China, Shanghai Aladdin), and mannitol (China, Shanghai Aladdin), were added separately to screen for the best supplementary carbon sources. Additionally, nitrogen sources at a concentration of 2% (*w*/*w*), such as peptone (China, Shanghai Macklin), yeast extract fermentation (YEF) (China, Shanghai Aladdin), NH_4_NO_3_, leucine (Leu), glycine (Gly), and arginine (Arg), were incorporated. For inorganic salts, 2% (*w*/*w*) NaCl, KCl, KH_2_PO_4_, CaCl_2_, FeSO_4_, and MgSO_4_ was added to the medium. The screening of these carbon sources, nitrogen sources, and inorganic salts was based on the yield of lunate conidia following the solid fermentation of Fxy121. Furthermore, the addition of glucose, YEF, and MgSO_4_ was tested at levels ranging from 2% to 14% (*w*/*w*) to identify the optimal addition amounts. Using an OED, three nutrients were set at three levels (Appendix A), and the best nutrient combination was selected based on the yield of lunate conidia.

### 2.4. Screening of the SSF Conditions and Response Surface Optimization

In this experiment, various environmental conditions were systematically evaluated for their impact on spore production. The temperature was varied between 23 °C and 27 °C, the humidity levels were adjusted to range from 50% to 90% RH, and inoculum levels of the spore suspension of 5%, 10%, 15%, 20%, 25%, and 30% (*v*/*w*) were tested, with the fermentation time set at 3, 5, 7, 9, and 11 days. The effects of each condition on the yield of lunate conidia were assessed to identify the optimal fermentation parameters.

Building on the single-factor experiments, a response surface methodology (RSM) was employed using the Box–Behnken design (BBD) principle with the software Design Expert 11. The yield of Fxy121 lunate conidia served as the response variable to assess the significance of the effect of each independent factor on the yield and to identify the optimal combination of components. The four factors examined were A (temperature, °C), B (humidity, % RH), C (inoculation volume, % (*v*/*w*)), and D (fermentation time, d), each categorized into three levels: −1, 0, and 1 (Table 1). The independent variables were coded according to Equations (1)–(4)(1)$$A=\frac{\left(\mathrm{Temperature}-25\right)}{1}$$(2)$$B=\frac{\left(\mathrm{Humidity}-80\right)}{10}$$(3)$$C=\frac{\left(\mathrm{Inoculation}\;\mathrm{volume}-15\right)}{5}$$(4)$$D=\frac{\left(\mathrm{Fermentation}\;\mathrm{time}-7\right)}{1}$$

The second-order polynomial coefficients were calculated and analyzed using Design Expert 11. A statistical analysis (analysis of variance) of the model was performed, which included Fisher’s F-test (overall model significance), its associated probability p (F), the correlation coefficient (R), and the determination coefficient (R^2^) to measure the goodness of fit of the regression model. The regression equation used to explain the influence of temperature, humidity, inoculation volume, and fermentation time on the lunate conidia production of Fxy121 is(5)$$Y=a0+a1A+a2B+a3C+a4D+a11A^{2}+a22B^{2}+a33C^{2}+a44D^{2}+a12A\times B+a13A\times C+a14A\times D+a23B\times C+a24B\times D+a34C\times D$$
where Y is the response calculated by the model, representing the yield of lunate conidia; a0 is the intercept; a1, a2, a3, and a4 are the linear effects; a11, a22, a33, and a44 are quadratic effects; a12, a13, a14, a23, a24, and a34 are the interaction coefficients; A is the temperature (°C); B is the humidity (% RH); C is the inoculation volume (% (*v*/*w*)); and D is the fermentation time (d).

Finally, response surface plots and contour plots were generated to investigate the correlation between the response variable and the independent variables.

### 2.5. Measurement of Lunate Conidia Production

After cultivating Fxy121 in solid fermentation medium for 9 days, 1 g of the contaminated medium was accurately weighed under sterile conditions and transferred to a 50 mL centrifuge tube. Next, 20 mL of a 0.5% Tween 80 solution was added, and the mixture was shaken at 25 °C and 200 r/min for 2 h. The lunate conidia produced in the fermentation medium were then counted using a hemocytometer, and the production was calculated using Equation (6)(6)$$\mathrm{Lunate}\;\mathrm{conidia}\;\mathrm{production}\;\left(\mathrm{conidia}/g\right)=\frac{\mathrm{Number}\;\mathrm{of}\;\mathrm{lunate}\;\mathrm{conidia}\;\mathrm{in}\;80\;\mathrm{small}\;\mathrm{squares}}{80}*400*10000*\mathrm{dilution}\;\mathrm{factor}$$

### 2.6. Data Analysis

To ensure the reliability of the results and minimize the impact of random errors, each treatment was repeated three times. After collecting the experimental data using the OED, range analysis (ANORA) and analysis of variance (ANOVA) were conducted. ANORA assumes that, when analyzing the effect of a particular factor, the influence of other factors on the results is balanced. ANOVA is used to estimate the percentage contribution of each parameter to the overall response, thereby assessing its relative significance. The RSM was implemented with Design Expert 11 software following the BBD. This analysis optimized the culture conditions across four factors at three levels. The corresponding results were obtained based on the optimal culture conditions predicted by the RSM. The ANOVA was conducted using IBM SPSS Statistics 23.0 (IBM Corporation). The results of the ANOVA were further analyzed through multiple comparisons using the least significant difference (LSD) method (*p* < 0.05).

## 3. Results

### 3.1. SSF Composite Medium for Fxy121

The yield of lunate conidia from Fxy121 in SSF was assessed using a hemocytometer, revealing an initial yield of 3.04 × 10^8^ conidia/g on the solid fermentation medium. A total of 25 experiments were designed and conducted utilizing an OED, with the yield of lunate conidia ranging from 0.93 × 10^8^ conidia/g to 7.63 × 10^8^ conidia/g (Appendix A). Subsequently, ANORA and ANOVA were performed on the experimental results. The ANORA revealed that wheat bran led to the largest range of 2.88, followed by the substrate-to-water ratio with a range of 2.45, while soybean flour led to the smallest range of 1.27 (Table 2). Consequently, the primary and secondary factors affecting the yield of Fxy121 lunate conidia were identified as follows: (primary) wheat bran → substrate-to-water ratio → corn flour → soybean flour (secondary). The maximum K values for each factor were $\bar{K3A}$ = 6.01, $\bar{K3B}$ = 5.4, $\bar{K2C}$ = 5.5, and $\bar{K3D}$ = 6.06, corresponding to the third level of A, the third level of B, the second level of C, and the third level of D, respectively. Thus, the optimal solid fermentation composite medium for maximizing the yield of Fxy121 lunate conidia involved a wheat bran/corn flour/soybean flour ratio of 7:2:1 (*w*/*w*/*w*), with a substrate-to-water ratio of 1:0.7 (*w*/*v*). Unfortunately, this combination was not included in the orthogonal experiments. Under these conditions, the yield of lunate conidia from the solid fermentation of Fxy121 was found to be 8.43 × 10^8^ conidia/g. This yield surpassed that of any combination tested in the orthogonal experiments, thereby confirming that these conditions represented the optimal combination of the examined parameters.

In the ANOVA, a larger F value indicated a stronger influence of each factor on the yield of Fxy121 conidia. In this experiment, the F value for wheat bran was the highest at 10.37, followed by the substrate-to-water ratio at 7.42, while the F value for soybean flour was the lowest at 2.47 (Table 3). This established the order of influence on the yield of Fxy121 conidia, i.e., wheat bran > substrate-to-water ratio > corn flour > soybean flour, which aligns with the order derived from the AVORA.

### 3.2. Screening and Optimization of Supplementary Nutrients

Different carbon sources, nitrogen sources, and inorganic salts were added to the medium to screen for the optimal supplementary nutrients for the culture medium. When glucose was used as the carbon source, the yield of Fxy121 lunate conidia reached its highest level at 10.24 × 10^8^ conidia/g. The addition of YEF as the nitrogen source resulted in a peak yield of 9.77 × 10^8^ Fxy121 lunate conidia. Additionally, when MgSO_4_ was included as the inorganic salt, the yield was the highest at 9.24 × 10^8^ conidia/g (Table 4). These yields of lunate conidia were significantly greater than those obtained with other carbon sources, nitrogen sources, and inorganic salts (*p* < 0.05).

We then added different concentrations of glucose, YEF, and MgSO_4_ to the medium. At a glucose concentration of 2% (*w*/*w*), the yield of Fxy121 lunate conidia was maximized at 11.83 × 10^8^ conidia/g. When the YEF concentration was 4% (*w*/*w*), the yield peaked at 11.64 × 10^8^ conidia/g. Similarly, a MgSO_4_ concentration of 4% (*w*/*w*) yielded the highest spore count at 10.48 × 10^8^ conidia/g (Table 5). These yields of lunate conidia were significantly higher than those obtained with other treatments (*p* < 0.05). Given that the added amounts of carbon sources, nitrogen sources, and inorganic salts should not be excessively high, the optimal total addition of glucose, YEF, and MgSO_4_ was determined to be 4%.

Using glucose, YEF, and MgSO_4_ as the three chosen factors, an OED was employed to design and conduct a total of nine experiments, with the yield of lunate conidia ranging from 8.55 × 10^8^ conidia/g to 13.89 × 10^8^ conidia/g (Appendix A). Subsequently, ANORA and ANOVA were performed on the experimental results. The ANORA revealed that glucose led to the largest range of 3.66, followed by YEF with a range of 1.48, while MgSO_4_ led to the smallest range of 0.9 (Table 6). Consequently, the primary and secondary factors affecting the yield of Fxy121 lunate conidia were identified as follows: (primary) glucose → YEF → MgSO_4_ (secondary). The maximum K values for each factor were $\bar{K1E}$ = 12.61, $\bar{K1F}$= 11.96, and $\bar{K1G}$ = 11.37, corresponding to the first level of E, the first level of F, and the first level of G, respectively. Thus, the optimal nutrient ratio for the solid fermentation medium of Fxy121 was glucose/yeast extract/magnesium sulfate = 1:1:1 (*w*/*w*/*w*), resulting in the highest yield of lunate conidia of 13.89 × 10^8^ conidia/g. Given that the total addition amount was 4% (*w*/*w*), the optimal concentrations in the Fxy121 solid fermentation medium were determined to be 1.33% (*w*/*w*) glucose, 1.33% (*w*/*w*) YEF, and 1.33% (*w*/*w*) MgSO_4_, which yielded the highest production of lunate conidia.

In the ANOVA, the F value for glucose was the highest at 135.99, followed by that of YEF at 26.86, and that of MgSO_4_, which was the smallest at 8.82 (Table 7). This confirmed the order of influence on the yield of lunate conidia as glucose > YEF > MgSO_4_, which is consistent with the order derived from the ANORA.

### 3.3. Screening of the SSF Conditions for Fxy121

When other conditions were held constant, temperature significantly affected the yield of lunate conidia of Fxy121, with the highest yield observed at an SSF temperature of 25 °C (Figure 1A). Similarly, humidity also had a significant impact on the yield, with the optimal yield occurring at an SSF RH of 80% (Figure 1B). Additionally, the inoculum volume played a crucial role, with the highest yield achieved at an inoculum volume of 15% (*v*/*w*) in the solid material (Figure 1C). Furthermore, the fermentation time significantly influenced the yield, with the maximum yield obtained after 7 days of fermentation (Figure 1D). The spore yields of Fxy121 reported above were significantly higher than those obtained with other treatments (*p* < 0.05).

### 3.4. Response Surface Analysis of the SSF Conditions for Fxy121

According to the BBD principle, the production of lunate conidia was selected as the response variable, with four factors, i.e., A (temperature), B (humidity), C (inoculation volume), and D (fermentation time), serving as independent variables in the central composite design. A total of 29 experimental runs were conducted. The results indicated that the production of lunate conidia under various SSF conditions ranged from 12.07 × 10^8^ conidia/g to 16.81 × 10^8^ conidia/g (Appendix A). The ANOVA results for the quadratic regression model of lunate conidia production indicated that the probability value of the regression model was *p* < 0.0001, and the lack-of-fit *p*-value was 0.7226 (>0.05), indicating that the regression model was highly significant and that the lack-of-fit was not significant (Table 8). This suggested minimal interference from unknown factors on the experimental results, confirming the stability of the model. The R^2^ value of the model was 96.73%, indicating that the model could explain 96.73% of the variation in spore production. The adjusted R^2^ value was 93.46%, further suggesting that the experimental values and the predicted values were in good agreement. This demonstrated that the generated model adequately explained the relationship between the lunate conidia production by Fxy121 and the fermentation temperature, humidity, inoculation volume, and fermentation time. Therefore, this model can be utilized for analysis and prediction. The impact of independent variables on the lunate conidia production of Fxy121 in SSF was assessed using Equation (7).(7)$$Y=16.38-0.3667A-0.2858B+0.1042C+0.1783D-2.09A^{2}-1.4B^{2}-0.2083C^{2}-0.972D^{2}-0.065A\times B-0.0175A\times C+0.0025A\times D-0.0375B\times C-0.005B\times D+0.0125C\times D$$

A larger F-value indicated a stronger effect of the factors on the lunate conidia production by Fxy121 (Table 8). The experimental results revealed that the influence of the four factors on lunate conidia production followed the order A (temperature) > B (humidity) > D (fermentation time) > C (inoculation volume). The significance test of the regression equation coefficients showed that temperature (A), humidity (B), and the quadratic terms (A^2^, B^2^, D^2^) had a highly significant impact on the lunate conidia production by Fxy121 (*p* < 0.01). *p*-values greater than 0.05 indicated that the effects of the model terms were not significant.

In this study, response surface plots and contour plots generated using Design Expert 11 were employed to investigate the effects of the interactions among the four key factors—temperature, humidity, inoculation volume, and fermentation time—on lunate conidia production. The response surface and contour plots for fermentation temperature and humidity (Figure 2A,B) indicated that, as the temperature increased from 24 °C to 26 °C and humidity rose from 70% RH to 90% RH, the yield of lunate conidia initially increased and then decreased. Similarly, the response surface and contour plots for fermentation temperature and inoculation volume (Figure 2C,D) demonstrated that, as the temperature rose from 24 °C to 26 °C and the inoculation volume increased from 10% (*v*/*w*) to 20% (*v*/*w*), the yield of lunate conidia also followed the same trend of first increasing and then decreasing. The response surface and contour plots for fermentation temperature and fermentation time (Figure 2E,F) revealed that, as the temperature increased from 24 °C to 26 °C and the fermentation time was extended from 6 days to 8 days, the yield of lunate conidia exhibited a similar pattern. Likewise, the response surface and contour plots for fermentation humidity and inoculation volume (Figure 2G,H) showed that, as humidity increased from 70% RH to 90% RH and the inoculation volume rose from 10% (*v*/*w*) to 20% (*v*/*w*), the yield of lunate conidia first increased and then decreased. The response surface and contour plots for fermentation humidity and fermentation time (Figure 2I,J) indicated that as humidity increased from 70% RH to 90% RH and the fermentation time was extended from 6 days to 8 days, the yield of lunate conidia followed the same trend. Additionally, the response surface and contour plots for inoculation volume and fermentation time (Figure 2K,L) demonstrates that, as the inoculation volume increases from 10% (*v*/*w*) to 20% (*v*/*w*) and the fermentation time was extended from 6 days to 8 days, the yield of lunate conidia also initially increased and then decreased. The slope of the response surface and the shape of the contour lines visually reflect the strength of the interactions between the factors; the more pronounced the interaction, the steeper the response surface and the more distinct the elliptical shape of the contour lines. Based on the contour plots, the interactions among these four factors were ranked as AB > BC > AC > CD > BD > AD, which aligns with the results of the ANOVA (Table 8). All response surface plots (Figure 2A–L) exhibited a peak point at the center, indicating that the lunate conidia production by Fxy121 was maximized when fermentation temperature, humidity, inoculation volume, and fermentation time were all maintained at moderate levels. According to the response surface optimization predictions made by the Design Expert 11 software, the optimal solid fermentation conditions for Fxy121 were a temperature of 24.9 °C, 79.0% RH, inoculation volume of 16.3% (*v*/*w*), and fermentation time of 7.1 days, predicting a maximum yield of lunate conidia of 16.44 × 10^8^ per gram of wet weight substrate containing Fxy121.

### 3.5. Validation of the Optimal Conditions

To validate the feasibility of the optimization results, we compared the yield of lunate conidia under the initial fermentation conditions with the yield obtained after response surface optimization. After 9 days of cultivation, the yield of lunate conidia for Fxy121 under the initial conditions was measured at 13.74 × 10^8^ conidia/g of wet weight substrate, while the yield after optimization increased to 16.58 × 10^8^ conidia/g. This result closely aligns with the predicted value from the response surface optimization, and the optimized lunate conidia yield was significantly higher than that observed under the initial fermentation conditions. Compared to the initial conditions, the lunate conidia yield of Fxy121 increased by 20.67%, further confirming the model’s feasibility.

## 4. Discussion

The fungus *E. vermicola*, with its lunate conidia capable of parasitizing PWN, has shown considerable efficacy in pest control. At present, *E. vermicola* is primarily cultivated in PDB liquid medium to achieve high yields [21]. While liquid fermentation can result in higher spore production, it lacks flexibility in terms of fungus storage and transportation [36]. In contrast, SSF offers advantages for the large-scale cultivation and transport of fungi. However, the specific formulation and optimal cultivating conditions for the SSF of *E. vermicola* to produce lunate conidia have not yet been clearly defined. This study aimed to improve the solid fermentation conditions for *E. vermicola* Fxy121 using an orthogonal experimental design and the response surface methodology to achieve the highest yield of lunate conidia, which is significant for future production processes and the development of biopesticides.

Agricultural industrial residues are often regarded as the most suitable substrates for SSF, including materials such as sugarcane bagasse, wheat bran, rice, wheat flour, corn flour, coffee grounds, apple pomace, soybean meal, and peanut cake [37]. Zhu et al. (2021) utilized rice as the solid fermentation substrate for *E. vermicola*, achieving a yield of lunate conidia of 1.88 × 10^8^ conidia/g after fermentation [32]. In contrast, under the same cultivation conditions, our study found that using wheat bran, corn flour, and soybean flour as substrates for Fxy121 resulted in a significantly higher yield, reaching 3.04 × 10^8^ conidia/g of lunate conidia. This suggests that this composite fermentation substrate is richer in nutritional components compared to single substrates, thereby promoting fungal germination and proliferation. The varying component ratios in the substrate have a significant impact on spore germination and reproduction. We determined that the optimal solid fermentation substrate for Fxy121 is a mixture of wheat bran, corn flour, and soybean flour in a ratio of 7:2:1 (*w*/*w*/*w*), resulting in a carbon-to-nitrogen (C/N) ratio of 35:1 (*w*/*w*), which allows for the highest production of lunate conidia for Fxy121. Du et al. (2013) reported that *E. vermicola* achieved maximum spore production with a C/N ratio of 40:1 (*w*/*w*) [38], which aligns with our findings. Furthermore, *E. vermicola* requires a certain amount of moisture during SSF. We found that a substrate-to-water ratio of 1:0.7 (*w*/*v*) resulted in higher lunate conidia production when using wheat bran, corn flour, and soybean flour as substrates, which is similar to the results of previous studies. Wang et al. (2013) used a substrate of wheat bran and pine wood powder to culture *E. vermicola*, with a substrate-to-water ratio of 1:1 (*w*/*v*) [26], and Zhu et al. (2021) cultured *E. vermicola* using rice as the substrate, with a substrate-to-water ratio of 2:1 (*w*/*v*) [32], indicating that the optimal substrate-to-water ratio in this study fell between 2:1 (*w*/*v*) and 1:1 (*w*/*v*).

Optimizing the basal fermentation medium is essential for microbial industrial production, serving as a critical pathway from laboratory research to industrial-scale applications [39,40]. Appropriate nutrient supplementation (i.e., carbon sources, nitrogen sources, and inorganic salts) can significantly enhance fungal conidial spore yields, reduce production costs, and have substantial implications for the field application of fungi in biological control. In this study, the optimal medium formulation for the SSF of Fxy121 to produce lunate conidia was determined to consist of wheat bran, corn flour, and soybean flour as a basal medium, supplemented with 1.33% (*w*/*w*) glucose, 1.33% (*w*/*w*) YEF, and 1.33% (*w*/*w*) MgSO_4_. Following medium optimization, the production of lunate conidia from Fxy121 significantly increased. Yang et al. (2016) found that adding 1.5% (*w*/*w*) glucose was beneficial for bacterial growth in the optimization of soybean meal SSF for producing nattokinase [41], which aligns with our findings. Nitrogen sources are essential nutrients for microbial growth and spore production, making the addition of nitrogen sources a crucial step to increase fungal spore yields. This study found that incorporating YEF into the solid medium promoted the production of lunate conidia by Fxy121. Jo et al. (2010) reported that adding yeast extract to the medium enhanced the mycelial proliferation and density of *Coriolus versicolor* [42], which is consistent with our results. In addition, readily available inorganic salts also play a significant role in mycelial growth and spore yield. Our study revealed that adding 1.33% (*w*/*w*) MgSO_4_ significantly promoted the production of lunate conidia by Fxy121. Similarly, Jo et al. (2009, 2010) found that incorporating MgSO_4_·7H_2_O into the medium positively affected the mycelial growth of *Coriolus versicolor* and *Ganoderma applanatum* [42,43]. Zhu (2021) reported that adding 1.775% (*w*/*w*) MgSO_4_ to the solid substrate could significantly increase the spore production of *E. vermicola* [44].

The fermentation of strains is influenced not only by the culture medium but also by the fermentation conditions, which are critical factors affecting the fermentation process [45]. This study identified the optimal fermentation conditions for the solid fermentation of Fxy121, which yielded the highest production of lunate conidia, as follows: a temperature of 24.9 °C, 78.0% RH, an inoculation volume of 16.3%, and a fermentation time of 7.1 days. The cultivation temperature is closely linked to microbial growth and is one of the most significant factors influencing microbial growth and reproduction. Excessively high temperatures can alter the membrane structure of microorganisms and degrade proteins, disrupting the activity of intracellular enzymes and severely inhibiting the normal microbial growth, which can lead to a substantial decrease in conidia yield [46]. Conversely, excessively low temperatures can slow the microbial growth and significantly prolong the time required for conidia production [47]. These findings are consistent with those of Wang et al., who reported that the optimal growth temperature for *E. vermicola* is between 25 and 26 °C, at which the growth rate is maximized, and the conidia germination rate is the highest [48]. Fungi can thrive at very low humidity levels if there is water present on their surface [49]. When only limited water is available, humidity plays a crucial role in mycelial growth [50]. In this study, Fxy121 produced the highest yield of lunate conidia in an environment with 78% RH, while a 50% RH environment significantly reduced the yield of lunate conidia and inhibited the strain growth. Research by Schoder et al. (2024) indicated that *Pleurotus ostreatus* and *Trametes versicolor* exhibited higher growth rates in an environment with 80% RH [51], which aligns with the results of this study. The inoculation volume is closely related to the reproduction of a strain; both insufficient and excessive inoculation volumes can adversely affect conidia production [52]. This study found that once the liquid inoculation volume reached the optimal level, further increases in the inoculation volume led to a significant reduction in the conidia yield of Fxy121. This may be attributed to the accumulation of secondary metabolites when the liquid inoculum is excessive, as an overabundance of metabolic products can negatively impact conidia production. The fermentation time significantly influences the moisture content, the nutrient availability of the substrate, and the age of an inoculum. Research by Ruijter et al. (2004) demonstrated that culture time and substrate moisture content have an antagonistic relationship [53], likely due to moisture consumption during the cultivation process. The solid fermentation cycle typically lasts between 3 and 10 days. Zhou et al. (2019) reported that the optimal fermentation cycle for *Paecilomyces lilacinus* is 7 days [54], and Yi et al. (2013) found that the optimal fermentation cycle for *Koningii trichoderma* is also 7 days [55].

## 5. Conclusions

This study successfully identified the optimal solid fermentation conditions for the strain Fxy121 through OED and response surface optimization, resulting in a significant increase in the yield of lunate conidia. The optimal culture medium was determined to be a mixture of wheat bran, corn flour, and soybean flour in a ratio of 7:2:1 (*w*/*w*/*w*), with a substrate-to-water ratio of 1:0.7 (*w*/*v*). Additionally, 1.3% (*w*/*w*) glucose, 1.3% (*w*/*w*) YEF, and 1.3% (*w*/*w*) MgSO_4_ were incorporated. The fermentation process was carried out at a temperature of 24.9 °C and 78.0% RH, with an inoculation volume of 16.3% (*v*/*w*) and a fermentation time of 7.1 days. This resulted in the highest yield of lunate conidia at 16.58 × 10^8^ conidia/g, representing a 4.45-fold increase compared to the pre-optimization yield of 3.04 × 10^8^ conidia/g. This achievement establishes a strong foundation for the industrial production of Fxy121 and has significant practical implications, as it can lower the production costs, enhance market competitiveness, and facilitate the application of green products such as biopesticides.

## Figures and Tables

**Figure 1 microorganisms-13-00434-f001:**
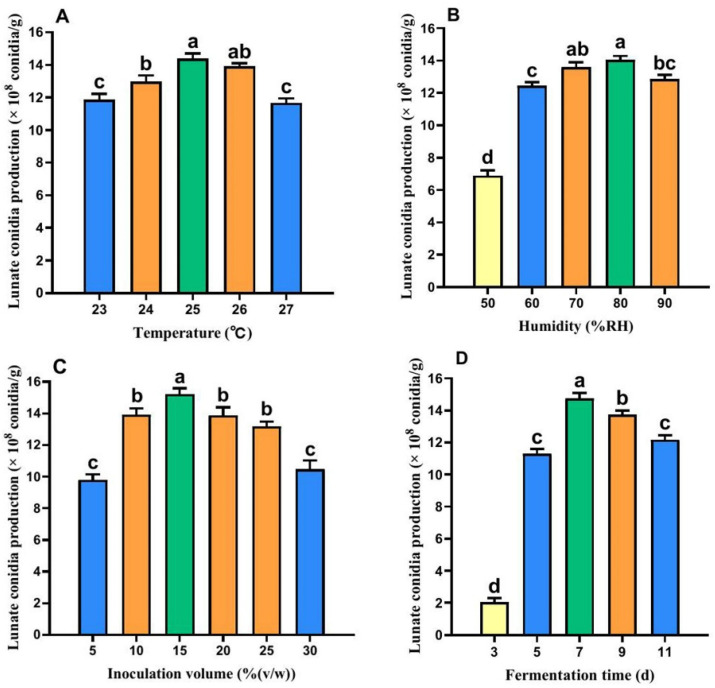
Effects of different SSF conditions on lunate conidia production by Fxy121. (**A**) The effect of fermentation temperature on the yield of lunate conidia of Fxy121. (**B**) The effect of fermentation humidity on the yield of lunate conidia of Fxy121. (**C**) The effect of Inoculation volume on the yield of lunate conidia of Fxy121. (**D**) The effect of fermentation time on the yield of lunate conidia of Fxy121. The lowercase letters indicate significant differences at the *p* < 0.05 level according to the LSD test.

**Figure 2 microorganisms-13-00434-f002:**
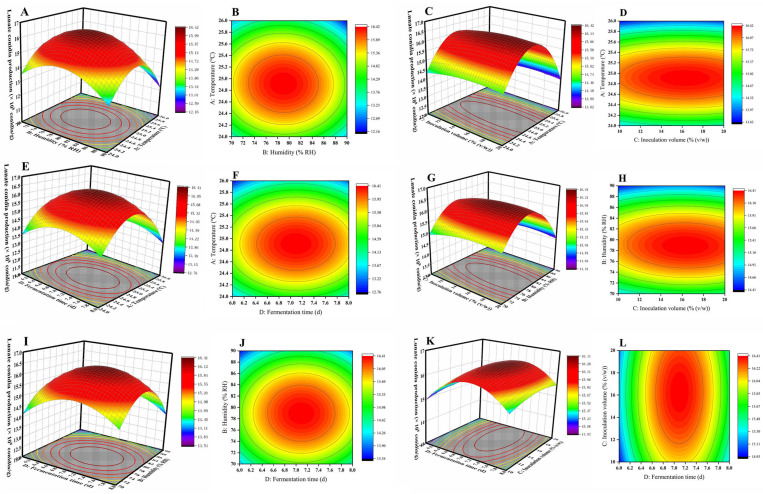
Response surface and contour plot for the SSF of Fxy121. (**A**,**B**) Response surface and contour plots of the effects of temperature and humidity on the yield of Fxy121 lunate conidia; (**C**,**D**) response surface and contour plots of the effects of temperature and inoculation volume on the yield of Fxy121 lunate conidia; (**E**,**F**) response surface and contour plots of the effects of temperature and fermentation time on the yield of Fxy121 lunate conidia; (**G**,**H**) response surface and contour plots of the effects of humidity and inoculation volume on the yield of Fxy121 lunate conidia; (**I**,**J**) response surface and contour plots of the effects of humidity and fermentation time on the yield of Fxy121 lunate conidia; (**K**,**L**) response surface and contour plots of the effects of inoculation volume and fermentation time on the yield of Fxy121 lunate conidia.

**Table 1 microorganisms-13-00434-t001:** The level of the independent variables used in response surface methodology (RSM).

Independent Variables	Unit	Code	Actual Levels of Coded Factor		
			−1	0	1
Temperature	°C	A	24	25	26
Humidity	% RH	B	70	80	90
Inoculation volume	% (*v*/*w*)	C	10	15	20
Fermentation time	d	D	6	7	8

**Table 2 microorganisms-13-00434-t002:** Range analysis (ANORA) of test results.

Index	A	B	C	D
	Wheat Bran	Corn Flour	Soybean Flour	Substrate-to-Water Ratio
K1	15.67	18.08	22.6	18.05
K2	24.34	25.77	27.48	27.09
K3	30.06	27	26.25	30.28
K4	27.31	26.57	25.82	25.12
K5	25.93	25.89	21.16	22.77
$\bar{K1}$	3.13	3.62	4.52	3.61
$\bar{K2}$	4.87	5.15	5.5	5.42
$\bar{K3}$	6.01	5.4	5.25	6.06
$\bar{K4}$	5.46	5.31	5.16	5.02
$\bar{K5}$	5.19	5.18	4.23	4.55
R	2.88	1.78	1.27	2.45
Ranking	A > D > B > C			
Optimal level	A_3_	B_3_	C_2_	D_3_

Note: Ki = sum of the production of lunate conidia at level i = 1, 2, 3, 4, 5 for the factor of interest (A, B, C, or D); $\bar{\mathrm{Ki}}$ = average of the production of lunate conidia at level i = 1, 2, 3, 4, 5 for the factor of interest (A, B, C, or D); $R=\bar{K_{\max}}-\bar{K_{\min}}$, where a larger R value indicates a more significant effect. The same notation applies hereinafter.

**Table 3 microorganisms-13-00434-t003:** Analysis of variance (ANOVA) of orthogonal test.

Factors	SS	df	MS	F	*p*	Significant
A	23.74	4	5.94	10.37	0.003	**
B	11.03	4	2.76	4.82	0.028	*
C	5.66	4	1.42	2.47	0.128	
D	16.99	4	4.25	7.42	0.008	**
Error	4.58	8	0.57			
Total	670.23	25				

Note: A: wheat bran; B: corn flour; C: soybean flour; D: substrate-to-water ratio; SS: sum of square; df: degree of freedom; MS: mean of square; F: F-value; **: *p* < 0.01; *: *p* < 0.05. The same notation applies hereinafter.

**Table 4 microorganisms-13-00434-t004:** The lunate conidia production by Fxy121 in solid-state fermentation (SSF) with different carbon sources, nitrogen sources, and inorganic salts.

Carbon Source	Lunate Conidia Production (×10^8^ Conidia/g)	Nitrogen Source	Lunate Conidia Production (×10^8^ Conidia/g)	Inorganic Salt	Lunate Conidia Production (×10^8^ Conidia/g)
Glucose	10.24 ± 1.12 a	Peptone	8.65 ± 1.38 ab	NaCl	3.78 ± 0.79 d
Sucrose	8.45 ± 1.02 ab	YEF	9.77 ± 1.61 a	KCl	5.41 ± 0.68 c
Lactose	8.59 ± 0.63 ab	NH_4_NO_3_	5.66 ± 0.85 c	KH_2_PO_4_	6.72 ± 0.38 b
Maltose	9.64 ± 0.72 a	Leu	7.64 ± 1.95 bc	CaCl_2_	4.67 ± 0.60 cd
Soluble starch	8.40 ± 1.07 ab	Gly	7.88 ± 0.80 abc	FeSO_4_	4.20 ± 0.64 cd
Mannitol	5.08 ± 1.00 c	Arg	7.87 ± 1.06 abc	MgSO_4_	9.24 ± 0.81 a
CK	7.33 ± 1.06 b	CK	7.33 ± 1.06 bc	CK	7.33 ± 1.06 b

Note: Data are presented as mean ± standard deviation. Lowercase letters indicate significant differences at the *p* < 0.05 level according to the least significant difference (LSD) test. The same notation applies hereinafter.

**Table 5 microorganisms-13-00434-t005:** The lunate conidia production by Fxy121 in SSF with different carbon source, nitrogen source, and inorganic salt addition amounts.

Addition Amount (% (*w*/*w*))	Lunate Conidia Production (×10^8^ Conidia/g)		
	Glucose	YEF	MgSO_4_
2	11.83 ± 0.89 a	9.17 ± 1.53 b	9.13 ± 0.25 b
4	9.41 ± 1.16 b	11.64 ± 0.50 a	10.48 ± 0.73 a
6	9.29 ± 0.65 b	9.14 ± 1.20 b	9.26 ± 0.59 b
8	7.49 ± 0.60 c	8.69 ± 0.62 b	8.68 ± 0.43 b
10	6.19 ± 0.46 d	6.79 ± 0.55 c	8.94 ± 0.69 b
12	6.10 ± 0.60 d	6.10 ± 0.80 cd	6.15 ± 0.51 c
14	5.00 ± 0.43 d	4.71 ± 0.39 d	4.63 ± 0.23 d

**Table 6 microorganisms-13-00434-t006:** ANORA of test results.

Index	E	F	G
	Glucose	YEF	MgSO_4_
K1	37.83	35.89	34.12
K2	34.36	31.45	33.52
K3	26.85	31.7	31.4
$\bar{K1}$	12.61	11.96	11.37
$\bar{K2}$	11.45	10.48	11.17
$\bar{K3}$	8.95	10.57	10.47
R	3.66	1.48	0.9
Ranking	E > F > G		
Optimal level	E_1_	F_1_	G_1_

**Table 7 microorganisms-13-00434-t007:** ANOVA of orthogonal test.

Factors	SS	df	MS	F	*p*	Significant
E	21	2	10.5	135.99	0.007	**
F	4.15	2	2.07	26.86	0.036	*
G	1.36	2	0.68	8.82	0.102	
Error	0.15	2	0.07			
Total	1116.54	9				

Note: E: glucose; F: YEF; G: MgSO_4_; **: *p* < 0.01; *: *p* < 0.05.

**Table 8 microorganisms-13-00434-t008:** ANOVA for the regression model.

Source	Sum of Squares	df	Mean Square	F-Value	*p*-Value
Model	40.75	14	2.91	29.58	<0.0001
A	1.61	1	1.61	16.39	0.0012
B	0.9804	1	0.9804	9.96	0.007
C	0.1302	1	0.1302	1.32	0.2693
D	0.3816	1	0.3816	3.88	0.0691
AB	0.0169	1	0.0169	0.1717	0.6849
AC	0.0012	1	0.0012	0.0124	0.9128
AD	0	1	0	0.0003	0.9875
BC	0.0056	1	0.0056	0.0572	0.8145
BD	0.0001	1	0.0001	0.001	0.975
CD	0.0006	1	0.0006	0.0064	0.9376
A^2^	28.46	1	28.46	289.13	<0.0001
B^2^	12.77	1	12.77	129.78	<0.0001
C^2^	0.2813	1	0.2813	2.86	0.113
D^2^	6.13	1	6.13	62.27	<0.0001
Residual	1.38	14	0.0984		
Lack of Fit	0.8637	10	0.0864	0.672	0.7226
Pure Error	0.5141	4	0.1285		
Total	42.13	28			
R^2^ = 96.73%	R^2^_Adj_ = 93.46%				

Note: A: temperature; B: humidity; C: inoculation volume; D: fermentation time.

## Data Availability

All data that support the findings of this study can be found within the paper and its Appendix A.

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
