# Peer review of "Screening and Optimization of Solid-State Fermentation for *Esteya vermicola*, an Entomopathogenic Fungus Against the Major Forest Pest Pine Wood Nematode"

_microorganisms, 2025, doi:10.3390/microorganisms13020434_

Round 1
Reviewer 1 Report
Comments and Suggestions for Authors
1. General Comments
The article is very well written. The conceptual development of the project is well founded, based on numerous bibliographic references. The work, very is interesting, contributes to the knowledge of forest management sustainability. Potentially, the research done can helps to develop new ways to control forest enemies.
2. Section by section
2.1. – Introduction
This section is very comprehensible, interesting and has recent bibliography to support the discussion made. The authors present a very detailed review about the subject explaining deeply the processes involved. For my point of view, a very good approach.
2.2. - Material and Methods
Material and Methods are easy to understand and allow to replicate the assay. I cannot assess whether the statistical methods are the most appropriate for the raised question; however, they seem to me very robust and allow for a convincing explanation of the obtained results.
2.3. - Results
Results are a little bit confusing.
I suggest changing the text from lines 240 to 250 and placing it below Table 3 to clearly separate the information from Table 3 and Table 4.
I do not understand why the paragraph of the current lines 248, 249, and 250 appears. I suggest review it and rewrite according with the text context.
Text referring to Table 6 should be inserted between Tables 5 and 6. In the text to be inserted, it should mention (Table 6).
2.4. Discussion
Discussion is well conducted. It opens the door to potential lines of investigation and application of the present study in pest control.
3. Other suggestions
There are missing spaces in the separation of paragraphs, such as between lines 371 and 372. I suggest a careful review of the final document to fill these gaps.
Author Response
Comment 1: I suggest changing the text from lines 240 to 250 and placing it below Table 3 to clearly separate the information from Table 3 and Table 4. Response 1: Thank you for pointing this out. We agree with this comment. Therefore, we have moved the text from lines 240 to 250 of the original manuscript to below Table 3 to clearly separate the information from Tables 3 and 4. The revised manuscript now has this text in lines 255 to 262. Comment 2: I do not understand why the paragraph of the current lines 248, 249, and 250 appears. I suggest review it and rewrite according with the text context. Response 2: Thank you for pointing this out. We agree with this comment. Therefore, we have deleted lines 248, 249, and 250 from the original manuscript. Comment 3: Text referring to Table 6 should be inserted between Tables 5 and 6. In the text to be inserted, it should mention (Table 6). Response 3: Thank you for pointing this out. We agree with this comment. Therefore, we have inserted the text referring to Table 6 above Table 6 and made the relevant explanations. Similarly, the explanation regarding Table 2 has also been adjusted. The revised manuscript now has this text in lines 230 to 235 and lines 285 to 288. Comment 4: There are missing spaces in the separation of paragraphs, such as between lines 371 and 372. I suggest a careful review of the final document to fill these gaps. Response 4: Thank you for pointing this out. We agree with this comment. Therefore, we have carefully reviewed the manuscript and added a space between lines 371 and 372 of the original manuscript. The revised manuscript now has this space in line 384.

Reviewer 2 Report
Comments and Suggestions for Authors
The paper is interesting, but it lacks key information. The experimental section needs further refinement. Comments are provided within the paper.

The English language needs improvement.
Author Response
Thank you very much for taking the time to review this manuscript. All revisions are highlighted in red in the manuscript. Below, I will provide a detailed explanation:
1) The term "Estaya vermicola" in the title has been italicized.
2) The phrase "Biocontrol Agent" in the title has been changed to "Entomopathogenic Fungi." This is because its crescent-shaped conidia can adhere to and kill the pine wood nematode, and the fungus has already been used in forests to prevent pine wilt disease. This is reflected in the Introduction, specifically in lines 50 to 64 of the revised manuscript.
3) In section 2.1 "Fungal strains and culture media," the full name of the laboratory and the name of the manufacturer of the Hemocytometer have been included in the text, located in lines 105, 106, and 114 of the revised manuscript.
4) In section 2.2 "Solid-state fermentation (SSF) culture," the initial conditions of the solid culture medium have been added, found in lines 119 and 120 of the revised manuscript.
5) The superscripts and subscripts in the text have been carefully checked and modified.

Round 2
Reviewer 2 Report
Comments and Suggestions for Authors
The paper can be accepted in this form.